# Effectiveness of an Intervention of Dietary Counseling for Overweight and Obese Pregnant Women in the Consumption of Sugars and Energy

**DOI:** 10.3390/nu11020385

**Published:** 2019-02-13

**Authors:** Elisa Anleu, Marcela Reyes, Marcela Araya B, Marcela Flores, Ricardo Uauy, María Luisa Garmendia

**Affiliations:** 1Institute of Nutrition and Food Technology (INTA), University of Chile, 7830490 Santiago, Chile; anleuelisa@gmail.com (E.A.); mreyes@inta.uchile.cl (M.R.); druauy@gmail.com (R.U.); 2Department of Women and Newborn Health Promotion, Faculty of Medicine, University of Chile, 8380453 Santiago, Chile; marbannout@uchile.cl; 3Corporación de Salud Municipal de Puente Alto, 8210269 Santiago, Chile; marcela.flores@cmpuentealto.cl; 4Department of Pediatrics, School of Medicine, Pontificia Universidad Católica de Chile, 8330023 Santiago, Chile

**Keywords:** nutritional intervention, pregnant women, overweight, obesity, total sugars, energy

## Abstract

Objective: Evaluate if an intervention based on nutritional counseling decreases total sugars and energy consumption in overweight and obese pregnant women, compared to their previous consumption and compared to women who only received routine counseling. Methods: Randomized study of two groups: dietary counseling (Intervention Group: IG) and routine counseling (Control Group: CG). The intervention consisted of three educational sessions focused on decreasing intake of foods that most contribute to sugars consumption. Changes in sugars and energy consumption were evaluated by a food frequency questionnaire before and after the intervention. Results: We evaluated 433 pregnant women, 272 in IG and 161 in CG, who before intervention had a mean consumption of 140 g total sugars and 2134 kcal energy per day. At the end of the intervention, the IG showed 15 g/day lower consumption of total sugars (95% CI: −25 and −5 g/day), 2% less total energy from sugars (95% CI: −3% and −1% g/day), and 125 kcal/day less energy than the CG (95% CI: −239 and −10 kcal/day). Table sugar, sweets, and soft drinks had the greatest reduction in consumption. Conclusions: The intervention focused on counseling on the decrease in consumption of the foods that most contribute to sugars consumption in overweight and obese pregnant women was effective in decreasing total sugars and energy consumption, mainly in the food groups high in sugars. Future studies should examine if this intervention has an effect on maternal and fetal outcomes.

## 1. Introduction

In recent years, Chile has had one of the highest prevalence rates of overweight and obesity in Latin America [1]. Overweight has 39% prevalence and obesity has 29%; 51% of women in reproductive age have a state of malnutrition due to excess [2]. This situation is of concern for the country, given the consequences it represents for maternal and infant health. Excessive weight during pregnancy and maternal obesity are associated with complications such as fetal trauma, congenital malformations, recurrent abortion, gestational hypertension, preeclampsia, macrosomia (birth weight >4 kg at birth), Caesarean births, and gestational diabetes, among others [3,4]. Long-term complications include the appearance of obesity and noncommunicable diseases in both mother and child.

The appearance of pregnancy complications depends on the pre-gestation nutritional status, but also on gestational weight gain (GWG). One of the most important contributors to GWG are eating habits. Elevated sugars consumption of foods with high energy value and high added sugars predisposes these women to excessive GWG [5,6]. The study by Renault et al. [6] revealed that large GWG is more related to intake of added sugars than to saturated fats. It is of interest to consider the nutritional state of pregnant women, emphasizing the habitual consumption of these foods.

Interventions focused on diet and physical exercise during pregnancy have shown to be efficacious in preventing excessive GWG and in other maternal–fetal outcomes, such as lower incidence of gestational diabetes and Caesarean births [7,8,9]. The dietary component of the interventions has usually been based on acquiring healthier eating habits (reduction of the glycemic index of foods, restricting the consumption of saturated fats, and increased consumption of fruits and vegetables [10,11,12]. Others have used recommendations based on food guides, considering preferences and alimentary beliefs [8]. However, there are no studies that show the effect of a nutritional intervention focused on the reduction in consumption of those foods that most contribute to total sugars consumption.

We developed a home intervention strategy in Chile based on diet counseling and omega-3 supplementation in order to decrease the incidence of gestational diabetes mellitus in women who began their pregnancy overweight or obese. The study protocol has been published elsewhere [13] (Trial registration: NCT02574767). Briefly, the intervention attempts to achieve adequate metabolic control for pregnant women and their children resulted in lower incidence of gestational diabetes mellitus and lower incidence of macrosomia in the newborn. The present study evaluates whether the dietary counseling decreased the consumption of total sugars and energy, compared to their consumption before the intervention and compared to women who only received routine counseling.

## 2. Materials and Methods

### 2.1. Study Design

The prospective experimental study evaluated the frequency of consumption before and after nutritional counseling of overweight and obese pregnant women, comparing the intervention group (IG) to a control group (CG). The IG received dietary counseling, while the CG received the routine control recommendations provided in the primary health centers of the Chilean Health-Care System.

### 2.2. Study Population

The study population consisted of pregnant women recruited from March 2016 to May 2018 in health centers of the Puente Alto county of Santiago, Chile: 272 participants in IG and 161 in CG. The inclusion criteria were: gestation of 15 weeks or less in the first prenatal control, 18 years of age or older, single pregnancy, and overweight or obese in the first control. The exclusion criteria were: previous diagnosis of diabetes or treatment with metformin or insulin, eating disorders (bulimia or anorexia), or a risk pregnancy defined in the health center using the definition in the Chilean Health Ministry guide [14].

The original group included 1002 women, 500 randomly assigned to IG and 502 to CG. In the present study we included the subsample of participants who answered the food frequency questionnaire (FFQ) at the beginning and end of the intervention between September 2016 and October 2018. Before the intervention, 408 and 264 participants answered the FFQ, respectively. At 35–37 weeks of pregnancy, 272 participants in the IG and 161 in the CG answered a second FFQ. The main reasons the second FFQs were not obtained are given in the flowchart in Figure 1. No significant differences were found in age, education, or nutritional status between the women who answered two questionnaires with those who answered only one, nor with the women who did not answer the FFQ (*p* > 0.05).

### 2.3. Dietary Counseling

The general objective of the dietary intervention is to reduce the consumption of the main sources of total sugars through a culturally tailored intervention culturally with recommendations simple and easy to follow by pregnant women. Nutritional intervention consisted of three teaching sessions focused on decreasing consumption of foods that are a relevant dietary source of sugars as well as the use of behavioral techniques to allow participants to achieve eating behavioral changes. There are no interventions similar to ours in the literature, but several clinical trials have also focused on decreasing the consumption of foods with high glycemic index [15,16,17,18]. The consumption of high carbohydrate foods influences maternal glucose levels, which results excessive gestational weight gain and fetoplacental growth [17]. Some interventions focused on reducing foods with high glycemic index have shown beneficial effects on maternal outcomes (gestational weight gain, fasting, and postprandial glucose levels) and offspring birth weight [19].

We previously used a multiple pass 24-h dietary recall (R24h) applied to 114 pregnant women who attended in the same health centers to identify the seven foods that most contributed to their sugars consumption (43.4% of total sugars consumption). The “Top 7” were sweetened soft drinks, juice with added sugar, powdered juices with sugar, cookies, table sugar, sweetened milk products, and bread. The first session took place when the participants had less than 15 weeks gestation, the second session at week 18, and the third between 24–28 weeks.

Session 1: “Introduction to gestational diabetes: sugars consumption during pregnancy and consequences for the baby”.

This consisted of showing an animated video with relevant information on gestational diabetes mellitus, illustrating the possible consequences of high sugars consumption on maternal–infant health and recommending general care in this phase of life, including healthy eating and physical activity. Participants were then given 6 photos with the Top 7 foods and a set of sugar cubes equivalent to 5 g each (one teaspoon). Participants were asked to place the sugar cubes that they thought were contained by each food on top of the corresponding photo. Then they were told the real amounts of sugar and given recommendations according to what each participant observed and commented.

Session 2: “Learning to substitute intelligently”.

This consisted of presenting information on options of healthy food substitutes for those with high sugars content. Each participant received a magnetic board with images of the Top 7, as well as a number of photos of healthy and unhealthy foods. The participants were asked to choose two alternatives that best substituted for the high-sugars foods in the same meal in which they eat those foods, and to place the two photos of each pair together. Afterwards, the selected choices for each food were analyzed, and recommendations to reinforce the consumption of healthy foods were given.

Session 3: “Identifying my eating habits”.

There were two activities in this session. The first activity used a board in the form of a traffic light; the participant was shown five photographs of eating habits of a pregnant woman, and a brief description of exactly what each photograph represented was read to her. The objective was for the participant to place the photograph on the color to which she considered it belonged, where green = healthy, yellow = caution, and red = risky.

The second activity was a roulette game with four colors: red, green, blue, and yellow. Each color represented one of the topics treated in the previous two sessions. The participant spun the wheel five times. According to the color that resulted for each time, she took a card from the stack with that color that had a question on the corresponding topic. There were direct questions and multiple-choice questions with three options, but only one correct.

According to the responses of the participant in the two activities her knowledge was reinforced, emphasizing her weak points, and a general feedback of all the educational sessions was given.

### 2.4. Routine Counseling

The CG only received the routine counseling given in primary health centers in prenatal controls.

### 2.5. Data Collection

When participants were enrolled, sociodemographic (age, marital status, occupation, income, composition of the home), obstetric (previous pregnancies and abortions, etc.), and morbidity (personal and family history of depression, type 2 diabetes, hypertension) information was collected, and blood pressure, height, and weight were measured. We also recorded information about the group of omega-3 fatty acids in which women were randomly allocated to receive (200 mg/day or 800 mg/day). By far, this information is not known to the participants or the researchers; thus, we classified this variable for the analyses as group 1 and group 2. GWG was calculated as the difference between the weight at the last visit during pregnancy (35–37 weeks) and pre-pregnancy weight; it was analyzed both as continuous (kg) and as categorical below, within and above the Institute of Medicine (IOM) 2009 guidelines (7–11.5 kg for overweight and 5–9 kg for obesity) [20].

Food consumption was measured using an FFQ applied twice by trained nutritionists. The first was applied when the participant was enrolled in the study (before 15 weeks of gestation) and the second was applied after the last visit (weeks 35–37 of gestation); both measured food consumption in the previous month.

The semi-quantitative FFQ was designed to evaluate the habitual intake of foods that were a relevant dietary source of total sugars and energy and, according to data derived from R24h mentioned above, in which all foods and culinary preparations eaten by the participant were compiled. One R24h was collected for each participant by a single trained dietitian using the multi-pass method [21] and a photographic atlas of standard portions of usual foods and culinary preparations [22,23]. Contents of energy and total sugars of foods were computed according to the weight/volume consumed and based on nutrient composition databases. In the case of natural foods, we used the Chilean database [24] and the United States Department of Agriculture [25] (USDA) nutrient database; in the case of packaged foods, we used a database previously built by our research group, based on nutrient fact panels of packaged foods collected from supermarkets during 2015 [26]. No specific software was used for dietary assessments. Food items consumed by participants were identified with numbers from 1 to 489 and then grouped into 171 food items according to their similarity in physical characteristics or the content of energy or sugars in 100 g of the foods. We then calculated the percentage that each food contributed to the total intake of energy and total sugars of the diet. These food items were listed in decreasing order according to their contribution and the accumulated percentage calculated. Food groups that provided 95% of the energy and total sugars were selected to design the FFQ. The R24H results were also used to estimate the portions of each food usually consumed; these were selected as the standard portions.

The final FFQ instrument contained 86 food items grouped in 16 food groups: cereals, tubers, bread and cakes, cookies, snacks, sugars and sweets, ice cream, soft drinks and juices, fruits, vegetables, legumes, meats, sausages, whole milk products, Chilean prepared foods, and oils. There were six fruits that were asked about in all months, and some that were asked about only during part of the year: an autumn–winter group from 21 March to 20 September and a spring–summer group from 21 September to 2 March.

To help visualize portions and aid in asking about consumption, there was a specific photographic atlas for this project with photographs of the standard portion of each food item, in the same order in which they were asked. The participant answered how much and how often she consumed this food in the month previous to the application of the FFQ.

### 2.6. Statistical Analysis

Energy and total sugars consumption were estimated by multiplying the consumption frequency of a food by the mass of the standard portion of the FFQ, giving the monthly intake in grams per food item. This was transformed to total daily intake in grams per item and then to daily energy and total sugars consumption using the nutritional information collected at the International Network for Food and Obesity/Non-Communicable Diseases Research, Monitoring and Action Support Project (INFORMAS) of the Instituto de Nutrición y Tecnología de los Alimentos (INTA) of the Universidad de Chile, and from the nutritional composition of foods of the USA Department of Agriculture (USDA).

The continuous variables were described by means and standard deviations, and the categories by frequencies and proportions.

Differences in the consumption of total sugars and energy between the IG and CG in the baseline and after the intervention were evaluated through the Student’s *t*-test.

The change in energy and total sugars consumption pre-and post-intervention within each group was evaluated using *t*-test for paired samples.

Finally, the effect of dietary counseling intervention (categorical predictor variable: 0 = no, 1 = yes) on total sugars consumption and energy (outcomes) was tested by multiple linear regression models adjusted for the baseline value and by covariates. We repeated the models considering the change between initial and final values in sugars/energy consumption as outcome.

All analyses were performed on an intention-to-treat basis, according to the treatment group allocated at randomization.

Statistical analyses were performed using the Stata program version 13 (StataCorp LP: College Station, TX, USA). The significance level for all tests was *p* < 0.05.

### 2.7. Ethical Aspects

The ethics committee of INTA of the Universidad de Chile approved the protocol and informed consent procedure of the study. The women studied accepted to participate by signing an informed consent form.

## 3. Results

### 3.1. Characterization of the Participants

The characteristics of the 433 women studied at time of recruitment are shown in Table 1. Mean age was 28 years (SD = 5.8); 63% were obese, no difference between groups. A total of 61% of the participants responded to the FFQ during autumn and winter. A percentage of 74% of the participants had finished secondary school and 55% were married; this percentage was higher in the IG than in the CG (*p* < 0.05). The percentage diagnosis of depression was 27% (*n* = 77), and not different between groups.

Mean GWG at 35–37 weeks was similar in both groups: 8.6 kg (SD = 4.6) in the IG and 8.7 kg (SD = 4.5) in the CG (*p* > 0.05). The proportion of women who exceeded the IOM recommendations for GWG did not differ between groups (36.2% in the IG and 34.0% in the CG, *p* > 0.05.

### 3.2. Change in Total Sugars and Energy Consumption between Groups

The total dietary changes in mean intake of total sugars and energy before and after the intervention and the differences between IG and CG are given in Table 2. The mean consumption of total sugars in the group intervened with counseling was 143 g/day before the intervention and 111 g/day after the intervention, with a significant reduction of −33 g/day (*p* < 0.05). There was also a significant reduction in total sugars consumption in the CG, from 137 g/day at the beginning to 124 g/day at the end (*p* < 0.05); however, the reduction in the IG was significantly greater (by 20 g) than in the CG (*p* < 0.05).

There was also a significant decrease in the total energy consumption in the IG, from 2203 to 1924 kcal/day, a decrease of 279 kcal/day. Total energy consumption also decreased in the CG, from 2064 kcal/day to 1996 kcal/day, but the difference was not significant (*p* = 0.191). The difference between the groups after the intervention (72 kcal/day) was not significant (*p* = 0.264). However, the reduction in energy consumption was significantly greater in the CG (*p* < 0.05).

### 3.3. Changes in the Consumption of Total Sugars and Energy by Food Group

Table 3 and Table 4 show the changes in mean consumption of total sugars and energy, respectively, in the food groups of the FFQ. There were significant decreases in total sugars consumption in the group with dietary counseling in soft drinks and juices, table sugar and sweets, fruits, and milk products, with decreases of 12, 9, 7, and 4 g/day, respectively (*p* < 0.05). Small but significant decreases in sugars consumption were also observed in tubers, ice cream, vegetables, prepared foods, and oils. The CG had significant decreases in sugars and sweets (−4 g/day) and soft drinks and juices (−9 g/day), and very slight decreases in vegetables, prepared foods, and oils. The decrease in consumption was significantly greater in the IG in sugars from fruit (−6 g/day), milk products (−3 g/day), and soft drinks and juices, and it showed a tendency to decrease in ice cream (−1 g/day, *p* = 0.08) and sugars and sweets (−4 g/day, *p* = 0.08). There was a significant reduction of total energy in the IG in the food groups in which the dietary consumption was focused; snacks decreased by 16 kcal/day, sugars and sweets by 44 kcal/day, soft drinks and juices by 50 kcal/day, and whole milk products by 52 kcal/day (*p* < 0.05). There were smaller but significant reductions in ice cream, tubers, fruits, vegetables, prepared meats, and oils (*p* < 0.05).

In the CG, there were significant decreases in the consumption of table sugar and sweets (−22 kcal/day), soft drinks and juices (−39 kcal/day) and a smaller decrease in the energy from vegetables (−3 kcal/day) (*p* < 0.05). Snacks, fruits, and whole milk products that showed greater decrease in the IG after counseling than in the CG, with −14 kcal/day, −30 kcal/day, and −42 kcal/day, respectively (*p* < 0.05).

### 3.4. Effect of the Nutritional Intervention on Consumption of Sugars and Energy

Table 5 shows the results of the linear and multiple regression models. The multiple model showed that the group with dietary counseling had a significant reduction in total sugars consumption of 15.23 g/day (95% CI −25.01 to −5.46 g/day; *p* = 0.002), in energy by 125 kcal/day (95% CI −239 to −10 kcal/day; *p* = 0.032), and in the percentage of energy derived from sugars by 2% (95% CI −3% to −1%; *p* = 0.002), compared to the control group.

In addition to baseline values, significant covariables in the multivariable models were maternal age (β coefficient = −1.1, 95% CI = −2.1, −0.1) and health care center (β coefficient= −14.7, 95% CI = −24.4, −4.9) for the model of sugars consumption, maternal age (β coefficient −16, 95% CI −27, −5), and GWG (β coefficient 18, 95% CI 6, 30) for the model of energy, and health care center (β coefficient −2, 95% CI −3, −1) for the model of percentage of energy derived from sugars.

We repeated the analysis using the change between initial and final consumption as the response variable in a sensitivity analysis. The multiple models had similar results (Appendix A).

## 4. Discussion

The present study showed that an intervention with dietary counseling in overweight or obese pregnant women was effective in decreasing the consumption of foods that most contribute to total sugars, energy, and the percentage of energy from sugars. Compared to participants who only received routine counseling, the IG reduced the mean consumption of total sugars by 15 g/day and energy by 125 kcal/day, and also reduced the percentage of energy from sugars by 2%.

We did not find reported nutritional interventions focused on decreasing sugars comparable to the present study. However, our results are consistent with other interventions of a similar nature. A study in Mexico determined if the nutritional status of overweight and obese pregnant women improved by implementing a personalized diet. These results indicated that the group of sugars had a greater decrease in consumption (15.1%) than the other food groups evaluated [5]. This could be attributed to the fact that the personalized diet emphasized eating foods with low glycemic index. Poston et al. provided diet counseling to obese pregnant women centered on consumption of foods with low glycemic index, including replacement of sugared soft drinks by ones with low glycemic index and reduction of saturated foods; they showed that intervened participants reduced mean intake of saturated fats, carbohydrates, and total energy, and increased intake of protein and fiber [10]. Petrella et al. [16] evaluated a nutritional intervention in overweight and obese pregnant women in which the primary focus of the dietary intervention was decreasing high-glycemic index foods consumption and substituting them with healthier alternatives. They observed that the group who received the intervention decreased the consumption of sugar and increased the consumption of vegetables and fruits [16]. Wolff et al. [27] performed a dietetic intervention restricted to energy consumption, providing a distribution of macronutrients according to Danish recommendations; they found changes in the composition of the diet and a decrease in total energy consumption similar to that found in our study [27]. In contrast, an Australian randomized LIMIT trial performed an intervention in lifestyle focused on maintaining equilibrium of carbohydrates, fats, and proteins to reduce intake of foods rich in refined carbohydrates and saturated fats. They did not find a significant difference in the mean daily energy consumption between the group of pregnant women randomized to the intervention and the control group [28]. Our intervention may have been more effective because it was focused on the foods that contribute most to sugars consumption, thus it may have been easier for the participants to follow the indications. It has been shown that dietary interventions during pregnancy are more effective than a combination of diet and physical activity in terms of weight gained during pregnancy, suggesting that there may be poorer compliance by participants when a large number of indications are given to the participants [29].

As far as we know, ours is the first Chilean study that presents detailed information on the consumption of energy and total sugars in different food groups of the diet of overweight and obese pregnant women. The dietary counseling was effective in differentially reducing some of the foods that most contribute to sugars consumption (the Top 7). Compared to women in the CG, participants from the IG had significant greater reductions of the intake (calories) of snacks and whole fat milk products, which was also the case for total sugars coming from whole fat milk products. Besides this, the intervention also decreased the intake of energy and sugars coming from fruits, even if fruits were not part of the targeted food groups. The intervention also decreased the consumption (calories) of sugars and sweets, and soft drinks and juices, but this was not significantly different from what happened in the CG. The intake of bread and cookies did not decrease. The decrease in more than 15 g of total sugars intake observed in the adjusted models corresponded to a decrease of 2% of energy from total sugars in the studied population. The baseline intake of total sugars was 26% of calories, which is much greater than the recommendations made for the general population (i.e., <20% of daily energy intake from total sugars) [30]. Thus, the reported intervention helped decrease one-third of the excessive intake of total sugars. Total sugars intake decreased mainly from whole fat milk products and fruits, which could be interpreted as an unexpected outcome. In the case of fruits, the intake of total sugars in the follow-up of the IG was 26 g, which corresponds to 2–3 servings of fruits, according to the recommendations (considering that most of them have about 10% of their weight as sugars). The women counseled also showed greater reduction (42 kcal/day) of intake of whole milk foods and increase of 18 kcal/day in low-fat dairy products. These results comply with the nutritional recommendations of the Health ministry for low-fat dairy products for pregnant women, considered essential during pregnancy [31]. Evidence has shown that the absorption of calcium increases during pregnancy, fulfilling the requirements of bone mineralization of the fetus [32]. The study of Osorio et al. [33] suggests that greater intake of low-fat dairy products is significantly associated with lower risk of gestational diabetes mellitus. These findings lend weight to the context of the present research [33].

The decrease in dairy products also implies a decrease in the consumption of both total and added sugars, since the amount of total sugars among all yogurts and most whole fat milks were 2–3 times the natural content of lactose (about 5% among milks and 7% among yogurts). Thus, the intervention also helped in decreasing the gap between the intake and the recommendation of added sugars [34] even when we cannot indicate in which extent given added sugars were not assessed.

As far as we know, it is not possible to anticipate the exact clinical impact of such dietary improvement, but given the current knowledge on the detrimental impact of an excessive intake of sugars [35] and the better understanding of the Developmental Origin of Disease paradigm [36,37], it is plausible that this changes would imply relevant improvements in women and offspring health. Although the evidence of the effect of dietary interventions in mothers with obesity focused on reducing foods with high glycemic index on maternal and offspring cardiometabolic health is not consistent and needs to be further studied, some animal experiments have shown positive results [19,38]. In our study, the intervention was not associated with differences in GWG or in the percentage of women who exceeded the IOM 2009 recommendations for GWG between groups; therefore, any effect of maternal diet on maternal and offspring outcomes could be not mediated by changes in GWG. However, it is possible that sample size of the present study is not large enough to ensure adequate power to detect differences on GWG between groups.

The strengths of this study include its cohort design with close follow-up of the participants and measurement of multiple variables, which helped fit the models. The construction of an FFQ aimed at the consumption of sugars and energy based on an R24H questionnaire in a population with similar characteristics allowed a reliable representation of daily consumption. FFQ is a method of dietary evaluation used in most epidemiological studies [39,40], considered to provide a valid model of long-term habitual consumption as well as being cheap, rapid, and easy to apply [41]. However, since FFQs are self-reported, they may be subject to report bias. The IG may have reported lower intake than the CG, given the fact that our intervention was not blind. However, the results of consumption in the baseline were similar to those reported in the R24h of women who did not participate in the intervention. Both groups showed decreases in the consumption of sugars and total energy, greater in the IG, although evidence shows that pregnant women tend to increase energy consumption during pregnancy [42]. Part of the decrease in the CG group may be due to the routine counseling received in the prenatal attention.

This study showed that focused dietary counseling provided in only three sessions during a 12-week period can be effective in decreasing the consumption of sugars and energy. This is an alternative to the current guidelines of the Chilean National Health Services, which recommend a generally healthy diet [31]. Future studies should evaluate if these changes produce maternal and child results (lower incidence of gestational diabetes and macrosomia, among others). It would also be interesting to know if these maternal dietary changes that may be motivated by the pregnancy are maintained after the women give birth.

## 5. Conclusions

The implementation of a dietary intervention with nutritional counseling focused on sugars consumption decreased the consumption of energy and total sugars in overweight and obese pregnant women, mainly in the food groups high in free sugars. Future studies should evaluate if these changes produce short- and long-term maternal and child results. Given its focus, this intervention may be replicable in all primary health centers for pregnant women with the highest nutritional risk.

## Figures and Tables

**Figure 1 nutrients-11-00385-f001:**
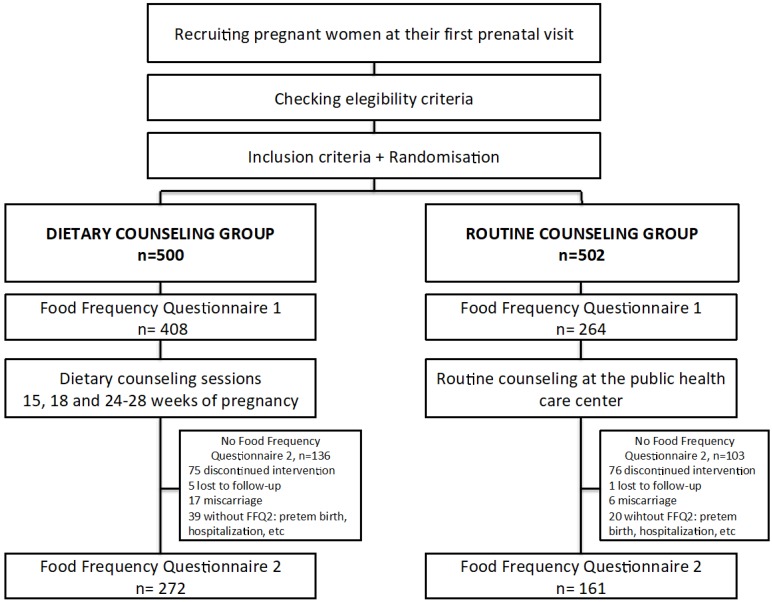
Flow diagram of the recruitment, randomization, and dietary measurements of trial participants.

**Table 1 nutrients-11-00385-t001:** Characterization of the study sample.

Variables	Control Group *N* = 161		Intervention Group *N* = 272		*p*-Value
	Mean (SD) *	N (%) **	Mean (SD) *	N (%) **	
*Age* (*years*)	28.2 (6.1)		27.9 (5.3)		0.629
*Nutritional status*					0.142
Overweight		67 (42)		94 (35)	
Obese		94 (58)		178 (65)	
*Marital state*					0.004
Single		82 (51)		100 (37)	
Married		79 (49)		172 (63)	
*Level of studies*					0.465
Primary school		33 (20)		70 (26)	
High school		119 (74)		188 (69)	
University		9 (6)		14 (5)	
*Activity*					0.449
Work		80 (50)		129 (47)	
Household labors		54 (33)		106 (39)	
Other		27 (17)		37 (14)	
*Number of persons in the home*	4.4 (1.9)		4.4 (1.7)		0.651
*Monthly household income* (*US dollars*)					0.333
≤ US 717		86 (53)		143(52)	
≥ US 717		64 (40)		99 (36)	
Other (no answer, doesn’t know)		11(7)		30 (11)	
*Relation to head of household*					0.229
Head of household		33 (21)		47 (17)	
Spouse, partner		57 (35)		119 (44)	
Other		71 (44)		106 (39)	
*Depression before pregnancy*					0.211
Yes		48 (30)		65 (24)	
No		111 (69)		206 (76)	
Doesn’t know		2 (1)		1 (0)	
*DHA supplementation*					0.968
Group 1		82 (51)		138 (51)	
Group 2		79 (49)		134 (49)	
*Health center*					
Low NSE		79 (49)		120 (44)	0.318
Medium NSE		82 (51)		152 (56)	

* Mean (standard deviation). ** *N* (%), Number (percentage).

**Table 2 nutrients-11-00385-t002:** Change in consumption of total sugars and energy.

	Control Group *N* = 161				Intervention Group *N* = 272				Intervention vs. Control					
	<15 Weeks of Pregnancy ^1^	35–37 Weeks of Pregnancy ^1^	Change ^1^	*p*-Value	15 Weeks of Pregnancy ^1^	35–37 Weeks of Pregnancy ^1^	Change ^1^	*p*-Value	Difference between Groups at < 15 Weeks Pregnancy ^2^	*p*-Value	Difference between Groups at 35–37 Weeks Pregnancy ^2^	*p*-Value	Difference in the Change ^2^	*p*-Value
Total sugars, g/day	136.94 (65.10)	123.94 (53.44)	−13.00 (60.11)	0.006	143.36 (70.33)	110.72 (55.01)	−32.64 (69.74)	<0.001	6.42 (6.80)	0.345	−13.22 (5.41)	0.015	−19.64 (6.59)	0.003
Total kilocalories, kcal/day	2064 (627)	1996 (627)	−68 (658)	0.191	2203 (794)	1924 (652)	−279 (761)	<0.001	139 (73)	0.058	−72 (64)	0.264	−211 (72)	0.003
Kilocalories of energy derived from sugars, kcal/day	548 (260)	496 (214)	−52 (240)	0.007	573 (281)	443 (220)	−130 (279)	<0.001	25 (27)	0.345	−53 (22)	0.015	−78 (26)	0.003
Percentage of energy derived from sugars, %	26 (8)	25 (6)	−1 (8)	0.016	26 (7)	23 (6)	−3 (8)	<0.001	0 (0)	0.519	−2 (1)	0.001	−2 (1)	0.060

Statistical differences were evaluated using Student’s *t*-test or *t*-test for paired samples. ^1^ Mean (standard deviation) ^2^ Mean difference (standard deviation).

**Table 3 nutrients-11-00385-t003:** Consumption of total sugars (g/day) by food groups.

	Control Group *N* = 161				Intervention Group *N* = 272				Intervention Group vs. Control Group					
Food Groups	<15 Weeks of Pregnancy ^1^	35–37 Weeks of Pregnancy ^1^	Change ^1^	*p*-Value	15 Weeks of Pregnancy ^1^	35–37 Weeks of Pregnancy ^1^	Change ^1^	*p*-Value	Difference between Groups at <15 Weeks of Pregnancy ^2^	*p*-Value	Difference between Groups at 35–37 Weeks of Pregnancy ^2^	*p*-Value	Difference in the Change ^2^	*p*-Value
Cereals	3.88 (4.77)	3.52 (3.68)	−0.35 (5.26)	0.393	3.35 (3.15)	2.97 (3.43)	−0.38 (3.79)	0.099	−0.53 (0.38)	0.161	−0.55 (0.35)	0.116	−0.03 (0.44)	0.952
Tubers	0.57 (0.42)	0.53 (0.44)	−0.04 (0.53)	0.349	0.62 (0.57)	0.52 (0.39)	−0.10 (0.57)	0.003	0.05 (0.05)	0.307	−0.01 (0.04)	0.768	−0.06 (0.05)	0.259
Bread	2.16 (1.81)	2.29 (2.14)	0.13 (2.38)	0.474	2.44 (2.03)	2.36 (1.76)	−0.8 (2.43)	0.594	0.28 (0.19)	0.139	0.07 (0.18)	0.742	−0.21 (0.24)	0.375
Pies and cakes	9.33 (8.08)	9.86 (9.12)	0.53 (10.63)	0.526	10.36 (10.73)	8.92 (8.69)	−1.44 (13.00)	0.068	1.12 (0.97)	0.250	−0.94 (0.88)	0.283	−1.97 (1.21)	0.103
Cookies	3.81 (6.83)	3.57 (4.90)	−0.24 (7.20)	0.672	3.70 (6.26)	3.06 (7.36)	−0.64 (8.09)	0.192	−0.05 (0.64)	0.931	−0.51 (0.65)	0.430	−0.40 (0.77)	0.602
Snacks	1.97 (2.73)	2.46 (3.64)	0.49 (4.36)	0.155	1.75 (2.79)	1.84 (3.43)	0.09 (4.19)	0.701	−0.23 (0.27)	0.394	−0.62 (0.35)	0.077	−0.39 (0.42)	0.353
Sugars and sweets	23.49 (18.69)	19.19 (18.50)	−4.30(20.57)	0.008	27.98 (24.68)	19.38 (19.18)	−8.60 (27.37)	<0.001	4.88 (2.23)	0.029	0.19 (1.88)	0.921	−4.30 (2.49)	0.084
Ice cream	2.28 (4.94)	2.65 (7.39)	0.36 (8.65)	0.595	2.71 (5.23)	1.83 (2.66)	−0.88 (5.95)	0.015	0.70 (0.45)	0.125	−0.82 (0.49)	0.096	−1.24 (0.70)	0.077
Soft drinks and juices	29.23 (35.80)	20.50 (23.96)	−8.73 (31.88)	<0.001	27.08 (32.45)	15.05 (20.51)	−12.03 (34.47)	<0.001	−2.15 (3.35)	0.521	−5.45 (2.17)	0.012	−3.30 (3.33)	0.323
Fruits	31.23 (23.16)	30.69 (23.77)	−0.54 (27.23)	0.799	33.03 (25.71)	26.41 (20.97)	−6.61 (25.29)	<0.001	1.89 (2.46)	0.445	−4.28 (2.18)	0.051	−6.07 (2.58)	0.019
Vegetables	1.79 (1.38)	1.49 (1.08)	−0.30 (1.35)	0.005	2.00 (1.73)	1.59 (1.27)	−0.41 (1.75)	<0.001	0.20 (0.16)	0.208	0.10 (0.12)	0.418	−0.11 (0.16)	0.478
Legumes	0.27 (0.30)	0.28 (0.30)	0.01(0.34)	0.691	0.26 (0.31)	0.27 (0.33)	0.01 (0.40)	0.572	−0.01 (0.03)	0.865	−0.01 (0.03)	0.776	0.00 (0.04)	0.934
Meats	0.29 (0.25)	0.28 (0.19)	−0.01 (0.29)	0.570	0.28 (0.25)	0.26 (0.24)	−0.02 (0.29)	0.295	−0.01 (0.02)	0.839	−0.01 (0.02)	0.579	−0.01 (0.03)	0.845
Sausage	0.20 (0.21)	0.18 (0.19)	−0.02 (0.23)	0.296	0.25 (0.24)	0.20 (0.21)	−0.05 (0.26)	0.002	0.05 (0.02)	0.039	0.02 (0.02)	0.408	-0.03 (0.02)	0.219
Whole milk products	11.67 (14.33)	10.67 (14.58)	−1.00 (15.11)	0.404	12.91 (16.02)	8.51 (13.32)	−4.40 (15.09)	<0.001	1.59 (1.51)	0.294	−2.16 (1.37)	0.115	−3.40 (1.49)	0.023
Low-fat milk products	12.23 (12.19)	14.18 (14.66)	1.95 (15.21)	0.103	10.00 (11.87)	14.19 (15.15)	4.19 (15.74)	<0.001	−2.30 (1.19)	0.054	0.01 (1.48)	0.995	2.24 (1.54)	0.148
Preparations	3.73 (3.68)	3.07 (3.04)	−0.66 (3.49)	0.019	4.54 (8.19)	3.27 (3.63)	−1.27 (7.67)	0.007	0.93 (0.68)	0.171	0.20 (0.34)	0.561	−0.62 (0.64)	0.334
Oils	0.06 (0.06)	0.04 (0.04)	−0.02 (0.06)	0.010	0.07 (0.08)	0.04 (0.04)	−0.03 (0.09)	<0.001	0.01 (0.01)	0.112	0.00 (0.00)	0.665	0.01 (0.01)	0.173

Statistical differences were evaluated using Student’s *t*-test or *t*-test for paired samples. ^1^ Mean (standard deviation) ^2^ Mean difference (standard deviation).

**Table 4 nutrients-11-00385-t004:** Energy consumption (kcal/day) by food group.

	Control Group *N* = 161				Intervention Group *N* = 272				Intervention Group vs. Control Group					
Food Groups	<15 Weeks of Pregnancy ^1^	35–37 Weeks of Pregnancy ^1^	Change ^1^	*p*-Value	15 Weeks of Pregnancy ^1^	35–37 Weeks of Pregnancy ^1^	Change ^1^	*p*-Value	Difference between Groups at < 15 Weeks of Pregnancy ^2^	*p*-Value	Difference between Groups at 35–37 Weeks of Pregnancy ^2^	*p*-Value	Difference in the Change ^2^	*p*-Value
Cereals	228 (119)	243 (125)	15 (139)	0.182	238 (132)	231 (143)	−7 (160)	0.527	10 (12)	0.452	-12 (13)	0.391	−21 (15)	0.170
Tubers	48 (35)	43 (34)	−5 (37)	0.070	54 (50)	44 (34)	−10 (50)	<0.001	6 (4)	0.193	1 (3)	0.845	−5 (4)	0.268
Bread	213 (179)	226 (210)	13 (234)	0.472	240 (200)	232 (174)	−8 (240)	0.595	28 (19)	0.139	6 (18)	0.741	−21 (23)	0.375
Pies and cakes	241 (207)	245 (215)	4 (236)	0.813	271 (226)	237 (229)	−34 (286)	0.053	30 (22)	0.172	−8 (22)	0.706	−38 (27)	0.154
Cookies	73 (102)	63 (73)	−10(110)	0.263	70 (102)	60 (114)	−10 (128)	0.208	−3 (10)	0.898	−2 (10)	0.813	0 (12)	0.999
Snacks	46 (51)	44 (49)	−2 (67)	0.785	51 (77)	35 (47)	−16 (79)	<0.001	5 (7)	0.428	−9 (5)	0.047	−14 (7)	0.047
Sugars and sweets	136 (114)	114 (100)	−22(121)	0.019	154 (126)	110 (98)	−44(144)	<0.001	20 (12)	0.087	−3 (9)	0.731	−21 (13)	0.117
Ice cream	22 (47)	25 (72)	3 (84)	0.621	26 (49)	17 (26)	−9 (56)	0.015	7 (4)	0.125	−8 (5)	0.115	−12 (7)	0.086
Soft drinks and juices	123 (151)	84 (104)	−39(135)	<0.001	112 (132)	62 (84)	−50 (140)	<0.001	−11 (14)	0.407	−22 (9)	0.014	−10 (14)	0.451
Fruits	169 (120)	165 (121)	−4(136)	0.692	177 (132)	143 (108)	−34 (130)	<0.001	8 (13)	0.516	−22 (11)	0.047	−30 (13)	0.022
Vegetables	16 (12)	13 (9)	−3 (11)	0.001	18 (15)	14 (11)	−4 (15)	<0.001	2 (1)	0.085	1 (1)	0.268	−1 (1)	0.348
Legumes	31 (31)	32 (30)	1 (37)	0.537	30 (32)	30 (33)	0 (39)	0.793	0 (3)	0.975	−2 (3)	0.520	−1 (4)	0.760
Meats	190 (99)	191 (85)	1 (107)	0.923	189 (100)	188 (97)	−1 (112)	0.891	−1 (9)	0.956	−3 (9)	0.763	−2 (11)	0.873
Sausage	42 (35)	39 (35)	−3 (41)	0.459	48 (41)	42 (38)	−6 (47)	0.021	6 (4)	0.089	2 (4)	0.534	−4 (4)	0.345
Whole milk products	143 (172)	133 (165)	−10 (160)	0.403	164 (171)	112 (135)	−52 (164)	<0.001	24 (17)	0.161	−20 (14)	0.162	−41 (16)	0.010
Low-fat milk products	129 (117)	141 (130)	12 (143)	0.287	115 (135)	145 (139)	30 (156)	0.001	−15 (13)	0.246	4 (13)	0.748	18 (15)	0.218
Preparations	152 (83)	143 (96)	−9 (104)	0.288	159 (106)	150 (96)	−9 (120)	0.236	8 (9)	0.379	7 (9)	0.442	0 (11)	0.994
Oils	75 (50)	69 (44)	−6 (55)	0.134	86 (68)	69 (43)	−17 (69)	<0.001	12 (6)	0.060	0 (4)	0.967	−11 (6)	0.094

Statistical differences were evaluated using Student’s *t*-test or t-test for paired samples. ^1^ Mean (standard deviation) ^2^ Mean difference (standard deviation).

**Table 5 nutrients-11-00385-t005:** Effect of the nutritional intervention on the consumption of total sugars and energy. Linear regression models.

	MODEL 1	*p*-Value	MODEL 2	*p*-Value	MODEL 3	*p*-Value
Total sugars consumption post intervention, g/day	−13.22 (−23.85, −2.57)	0.015	−15.44 (−25.04, −5.84)	0.002	−15.23 (−25.01, −5.46)	0.002
Total kilocalorie consumption post intervention, kcal/day	−71 (−197, 54)	0.265	127 (−239, −14)	0.028	−125 (−239, −10)	0.032
Total percentage of energy consumption derived from sugars, %	−2 (−3, −1)	0.001	−2 (−3, −1)	0.001	−2 (−3, −1)	0.002

Reference group is the control group. Figures represent β coefficients of the regression models and 95% confidence intervals. MODEL 1: Sugars/kilocalories/% calories from sugars at 35–37 weeks of pregnancy = β0 + β1 (dietary counseling). MODEL 2: Sugars/kilocalories/% calories from sugars at 35–37 weeks of pregnancy = β0 + β1 (dietary counseling) + β2 (sugars/kilocalories/% calories at < 15 weeks of pregnancy). MODEL 3: Sugars/kilocalories/% calories from sugars at 35–37 weeks of pregnancy = β0 + β1 (dietary counseling) + β2 (sugars/kilocalories/% calories at < 15 weeks of pregnancy) + β3 (marital state) + β4 (persons in home) + β5 (age) + β6 (level of studies) + β7 (monthly income) + β8 (relation to head of household) + β9 (activity) + β10 (depression) + β11 (nutritional status) + β12 (gestational weight gain) + β13 (season) + β14 (health center).

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
