# Peer review of "Effectiveness of an Intervention of Dietary Counseling for Overweight and Obese Pregnant Women in the Consumption of Sugars and Energy"

_nutrients, 2019, doi:10.3390/nu11020385_

Reviewer 1 Report

In this study, the authors tried to see if the implementation of a dietary intervention with a nutritional counseling focused to reduce total sugar in overweight/obese pregnant women, effectively reduces sugars consumption and energy intake, by using a list of “Top 7” foods.  However, there are some concerns need to be addressed.

 Main concerns:

In general, methods section is confusing, namely regarding participants’ selection. Please add a flow-chart of the study and clarify drop-out.

 Additional references are needed in the Dietary Counseling section.

 The authors use the term “Routine counseling” and also “ non-counseled” for the control group. It is nonsense.  

 The Statistical Analysis section (page 4) is highly disorganized and hard to follow. Please, rewrite, removing the sentences regarding chi square test and paired samples t test from the final part of this section to the initial part.

Please, specify better where nutritional data were collected (Did the author use a nutritional software? which database? reference period …years of this database).

It is not clear how the linear regression analysis was performed  (specify dependent and independent variables, how the models were performed).Since the intervention group has basal intake higher than control group,  did the authors expect an effect of this variable on some models?Are you sure, you do not need a logistic regression analysis?

Discussion is not specific enough for providing the reader with the information to this particular study. The authors should explain the clinical significance to reduce of 15 g/day of sugars in pregnancy.

 Did the author found body weight, glucose or lipids reduction?

 Please improve the discussion of the results.

Why the “Top 7” foods method was used? (add appropriate references).  Is this the first description for this method? Is it validated?

The authors described the merits of reducing total sugars intake but  table 3 shows a significant reduction in fruit and whole milk products.  Were these the goals to be achieved in this study? Why the reduction in  the intake of fruit and whole milks products is positive? Please, discuss  this finding.

Minor concerns

The term “Intervened Group”, please check.

Author Response

Response to Reviewer 1 Comments

Dear Reviewer 1:

We would like to thank you for the helpful comments. We have responded to each comment and suggestion individually on the pages that follow. We are hopeful that this careful revision will be satisfactory for the reviewers and the editorial team.

 Comments and Suggestions for Authors

In this study, the authors tried to see if the implementation of a dietary intervention with a nutritional counseling focused to reduce total sugar in overweight/obese pregnant women, effectively reduces sugars consumption and energy intake, by using a list of “Top 7” foods.  However, there are some concerns need to be addressed. 

Main concerns:

Point 1: In general, methods section is confusing, namely regarding participants’ selection. Please add a flow-chart of the study and clarify drop-out.

Response 1: We have added a flow diagram (see Figure 1) and clarify the participants´ selection in Methods section

Point 2: Additional references are needed in the Dietary Counseling section. 

Response 2: in this version we included more references.

Point 3: The authors use the term “Routine counseling” and also “non-counseled” for the control group. It is nonsense.  

Response 3: In this version we used only the term “Routine Counseling” for the control group.

Point 4: The Statistical Analysis section (page 4) is highly disorganized and hard to follow. Please, rewrite, removing the sentences regarding chi square test and paired samples t test from the final part of this section to the initial part. 

Response 4: We have rewritten the statistical analysis section.

Point 5: Please, specify better where nutritional data were collected (Did the author use a nutritional software? which database? reference period …years of this database). 

Response 5: We have clarified the process of data collection of diet.

 Point 6:It is not clear how the linear regression analysis was performed (specify dependent 

and independent variables, how the models were performed).

 Response 6: We think that the new version of the statistical analysis section is written in a more clear way.

 Point 7: Since the intervention group has basal intake higher than control group, did the 

authors expect an effect of this variable on some models? Are you sure, you do not need 

a logistic regression analysis?

 Response 7: The regression models included basal intakes of sugar/energy as covariables; additionally in this version we added a table (S1) considering the change in sugar/energy consumption as outcome. We used linear regression models instead of logistic regression models because the outcome variables (sugar/energy consumption) are continuous.

 Point 8: Discussion is not specific enough for providing the reader with the information to this particular study. The authors should explain the clinical significance to reduce of 15 g/day of sugars in pregnancy. 

 Response 8: We have included in the discussion a paragraph with the potential benefits of this reduction in sugars.

Point 9: Did the author found body weight, glucose or lipids reduction? 

Response 9: The clinical trial aimed to evaluate if a dietary counselling intervention and/or omega-3 supplementation decreases the incidence of gestational diabetes mellitus and lower incidence of macrosomia in the newborn in women who began their pregnancy overweight or obese. The objective of the present paper evaluates whether the dietary counseling decreases the consumption of total sugars and energy, compared to their consumption before the intervention and to women who only received routine counseling. Most of the existing literature related to dietary interventions during pregnancy in overweight and obese pregnant women reports the effects of the intervention on gestational weight gain or maternal and offspring outcomes. However, dietary changes following the interventions are poorly reported; therefore, it is difficult to know if the effect of the interventions on outcomes is mediated by dietary changes or by other factors. In this paper, we aimed to address maternal dietary changes following the dietary counseling intervention. Currently, all participants were recruited and counseling visits have been completed. However, the follow-up of the pregnant women until delivery has not ended, so these preliminary results are not included since they are beyond the scope of the proposed objective.

Point 10: Please improve the discussion of the results. 

Response 10: In this new version several paragraphs of the discussion were improved and more references were added.

Point 11: Why the “Top 7” foods method was used? (add appropriate references). Is this the first description for this method? Is it validated?

Response 11: In this version we clarified the overall goal of the dietary counselling and added references.

 Point 11:The authors described the merits of reducing total sugars intake but table 3 shows 

a significant reduction in fruit and whole milk products. Were these the goals to be 

achieved in this study? Why the reduction in the intake of fruit and whole milks products 

is positive? Please, discuss this finding.

Response 12: in the present version of the document, there is a deeper discussion of the results of the intervention including the unexpected result of the decrease in dairy products and fruits.

Minor concerns

Point 12: The term “Intervened Group”, please check.

Response 12: We have changed this term.

Reviewer 2 Report

Anleu et al. present results from an intervention study in pregnant women, aiming at decreasing sugar consumption and energy intake. The authors show that the counseling intervention significantly decreased intake of sugars and calories compared to the control group. The manuscript is well written and well presented. However, there several questions and concerns:

Major points

The authors show that the intervention reduced sugar and caloric intake. However, there is no data that links this improvement in nutrition to maternal or infant health. The authors state that further studies will be needed to address this point. While this reviewer understands that infant follow-up is complex and time consuming, materna data should be readily available. Providing maternal parameters (gestational weight gain, gestation diabetes, glucose homeostasis variables) would greatly increase the significance of the study. Does the change in energy and sugar consumption have an impact on GWG or diabetes? 

In the Methods section, the authors state that the participants were randomly selected from the original group amongst women that answered the FFQ at both time points. Can the authors clarify how this random selection of participants was performed? If it was random, why is there an unbalance in the number of women in each group?

Minor points

Tables 2, 3 and 4. While providing the paired comparison within the two groups before and after the intervention, this reviewer believes that the most important data is the comparison between both groups. These data are shown in the last column of the tables ("Difference in change); however, only the mean difference is shown with no SD or p value (although an asterisk marks the significance). Could the authors provide SD and p val for all these tables?

Related to the previous point, Table 5 should show the regressions using the change between initial and final values in addition to (or instead of) the final values only (the authors state these data are not shown at the end of the results section). 

The authors apply t-test and paired t-test to compare groups. These test require that data have a normal distribution. Have the authors tested normality in their dataset? Are all variables normally distributed?

Author Response

Response to Reviewer 2 Comments

 Dear Reviewer 2:

We would like to thank you for the helpful comments. We have responded to each comment and suggestion individually on the pages that follow. We are hopeful that this careful revision will be satisfactory for the reviewers and the editorial team.

 Comments and Suggestions for Authors

Anleu et al. present results from an intervention study in pregnant women, aiming at decreasing sugar consumption and energy intake. The authors show that the counseling intervention significantly decreased intake of sugars and calories compared to the control group. The manuscript is well written and well presented. However, there several questions and concerns:

Major points

Point 1: The authors show that the intervention reduced sugar and caloric intake. However, there is no data that links this improvement in nutrition to maternal or infant health. The authors state that further studies will be needed to address this point. While this reviewer understands that infant follow-up is complex and time consuming, maternal data should be readily available. Providing maternal parameters (gestational weight gain, gestation diabetes, glucose homeostasis variables) would greatly increase the significance of the study. Does the change in energy and sugar consumption have an impact on GWG or diabetes? 

Response 1: The clinical trial aimed to evaluate if a dietary counselling intervention and/or omega-3 supplementation decreases the incidence of gestational diabetes mellitus and lower incidence of macrosomia in the newborn in women who began their pregnancy overweight or obese. The objective of the present paper evaluates whether the dietary counseling decreases the consumption of total sugars and energy, compared to their consumption before the intervention and to women who only received routine counseling. Most of the existing literature related to dietary interventions during pregnancy in overweight and obese pregnant women reports the effects of the intervention on gestational weight gain or maternal and offspring outcomes. However, dietary changes following the interventions are poorly reported; therefore, it is difficult to know if the effect of the interventions on outcomes is mediated by dietary changes or by other factors. In this paper, we aimed to address maternal dietary changes following the dietary counseling intervention. Currently, all participants were recruited and counseling visits have been completed. However, the follow-up of the pregnant women until delivery has not ended, so these preliminary results are not included since they are beyond the scope of the proposed objective.

Point 2: In the Methods section, the authors state that the participants were randomly selected from the original group amongst women that answered the FFQ at both time points. Can the authors clarify how this random selection of participants was performed? If it was random, why is there an unbalance in the number of women in each group?

Response 2: We have clarified the participants´ selection and included a flow diagram (See Figure 1).

Minor points

Point 3: Tables 2, 3 and 4. While providing the paired comparison within the two groups before and after the intervention, this reviewer believes that the most important data is the comparison between both groups. These data are shown in the last column of the tables ("Difference in change); however, only the mean difference is shown with no SD or p value (although an asterisk marks the significance). Could the authors provide SD and p value for all these tables?

Response 3: We have included the mean difference with corresponding SD and p-value.

Point 4: Related to the previous point, Table 5 should show the regressions using the change between initial and final values in addition to (or instead of) the final values only (the authors state these data are not shown at the end of the results section). 

Response 4: We have included a supplementary table with the regression models considering as outcome the change in sugars/energy consumption (please see Table S1)

Point 5: The authors apply t-test and paired t-test to compare groups. These test require that data have a normal distribution. Have the authors tested normality in their dataset? Are all variables normally distributed?

Response 5: We have examined whether the dietary variables have a normal distribution or not and several of them do not. We repeated the differences between groups using non-parametric tests for matched and unmatched samples, Wilcoxon signed-rank test and Wilcoxon rank-sum (Mann–Whitney), respectively. The resulting p-values are very similar to those obtained with the t-test (see table below), so we prefer to keep the tables of the original version.

CONTROL GROUP

n=161

INTERVENTION GROUP

n=272

INTERVENTION VS. CONTROL

<15 weeks

of pregnancy1

35-37 weeks

of pregnancy1

Change1

p-value

15 weeks

of pregnancy 1

35-37 weeks

of pregnancy 1

Change1

p-value

p-value  of the

difference between

groups

at<15 weeks

pregnancy 2

p-value of the

difference

between groups

at 35-37 weeks

pregnancy

p-value of the

difference

in the change

Total sugars, g/day

120.71 (93.20, 159.04)

111.68 (85.52, 150.93)

-8.44 (-42.86,30.75)

0.146

132.45 (92.22, 176.19)

100.41 (77.34-136.68)

-24.27 (-75.44, 11.34)

<0.001< span="">

0.566

0.008

0.001

Total kilocalories,   kcal/day

2006 (1682, 2478)

1871 (1614, 2324)

-117 (-544, 276 )

0.123

2051 (1721, 2566)

1790 (1479, 2293)

-292 (-738, 284)

<0.001< span="">

0.176

0.153

0.017

Kilocalories of energy   derived from sugars, kcal/day

470 (357, 626)

413 (314, 549)

-34 (-171, 123)

0.011

494 (343, 667)

376 (272, 491)

-97 (-302, 45)

<0.001< span="">

0.679

0.021

0.001

Percentage of energy   derived from sugars, %

26 (21, 30)

24 (21, 28)

-0.4 (-5, 3)

0.164

25 (21, 31)

22 (19, 26)

-2 (-9, 2)

<0.001< span="">

0.333

0.004

0.025

Statistical differences were evaluated using Wilcoxon signed-rank test and Wilcoxon rank-sum (Mann–Whitney).

1 Median (p25-p75)

 Round  2

Reviewer 1 Report

No futher comments 

Author Response

Response to Reviewer 1 Comments

Dear Reviewer 1:

We would like to thank the reviewer 1 for his detailed comments and suggestions for the manuscript. We believe that the comments have identified important areas, which required improvement. The revised manuscript has benefited from an improvement in the overall presentation and clarity.

Point 1: No further comments

  Reviewer 2 Report

The authors have done a good job modifying the manuscript and answered most the points raised in first revision. However, there are still some concerns regarding the point 1. The authors justified not including maternal physiologic data (gestational weight gain, gestational diabetes, glucose/insulin, lipids) stating that "the objective of the present paper evaluates whether the dietary counseling decreases the consumption of total sugars and energy [...]". While this objective is accomplished in the manuscript, the addition of the maternal variables (i.e. assessing whether the change in sugar/energy intake is associated with changes in GWG or other variables) would greatly increase the significance of this study. 

The authors further justify the non inclusion of these data stating that "the follow-up of the pregnant women until delivery has not ended [...]". This sentence is a bit confusing to this reviewer... Since the authors have analyzed the food questionnaires at 35-37 weeks of gestation, the follow-up until delivery of the participants included in this manuscript should be finished, and data for GWG or gestational diabetes should be readily available. Indeed, GWG has been included as a covariate in Model 3 of Table 5, but changes in this variable with the intervention compared to the control group are not shown. Does GWG change between groups? Is there any association between GWG and change in sugar/energy intake?

Author Response

  Response to Reviewer 2 Comments

Dear Reviewer 2:

We would like to thank the reviewer 1 for his detailed comments and suggestions for the manuscript. We believe that the comments have identified important areas, which required improvement. The revised manuscript has benefited from an improvement in the overall presentation and clarity.

Point 1: The authors have done a good job modifying the manuscript and answered most the points raised in first revision. However, there are still some concerns regarding the point 1. The authors justified not including maternal physiologic data (gestational weight gain, gestational diabetes, glucose/insulin, lipids) stating that "the objective of the present paper evaluates whether the dietary counseling decreases the consumption of total sugars and energy [...]". While this objective is accomplished in the manuscript, the addition of the maternal variables (i.e. assessing whether the change in sugar/energy intake is associated with changes in GWG or other variables) would greatly increase the significance of this study. 

The authors further justify the non-inclusion of these data stating that "the follow-up of the pregnant women until delivery has not ended [...]". This sentence is a bit confusing to this reviewer... Since the authors have analyzed the food questionnaires at 35-37 weeks of gestation, the follow-up until delivery of the participants included in this manuscript should be finished, and data for GWG or gestational diabetes should be readily available. Indeed, GWG has been included as a covariate in Model 3 of Table 5, but changes in this variable with the intervention compared to the control group are not shown. Does GWG change between groups? Is there any association between GWG and change in sugar/energy intake?

Response 2: This study included the subsample of participants who answered the food frequency questionnaire (FFQ) at the beginning and end of the intervention between September 2016 and October 2018 (433 from a total sample of 1000 participants). The follow-up until delivery of the 443 participants has ended and we have gestational weight gain (GWG) data from them, but not from all the participants because 8% of them have not yet finished their pregnancy. Our concern is that sample size of the present study could be not large enough to ensure adequate power to detect differences on outcomes between groups. Nevertheless, we agree with the reviewer that this information can improve the scope of the document. In this version, we have included data of GWG in both groups of randomization, and the role of GWG in the model between the intervention and the study outcomes. Please see methods, results and discussion where this information was added.